# Sparking Religious Conversion through AI?

**Moira McQueen**

University of St. Michael's College, University of Toronto, Toronto, ON M5S 1J4, Canada;
moira.mcqueen@utoronto.ca

**Abstract:** This paper will take the stance that cognitive enhancement promised by the use of AI could be a first step for some in bringing about moral enhancement. It will take a further step in questioning whether moral enhancement using AI could lead to moral and or religious conversion, i.e., a change in direction or behaviour reflecting changed thinking about moral or religious convictions and purpose in life. One challenge is that improved cognition leading to better moral thinking is not always sufficient to motivate a person towards the change in behaviour demanded. While some think moral bioenhancement should be imposed if necessary in urgent situations, most religions today see volition in conversion as essential. Moral and religious conversion should be voluntary and not imposed, and recent studies that show possible dangers of the use of AI here will be discussed along with a recommendation that there be regulatory requirements to counteract manipulation. It is, however, recognized that a change in moral thinking is usually a necessary step in the process of conversion and this paper concludes that voluntary, safe use of AI to help bring that about would be ethically acceptable.

**Keywords:** cognitive and moral enhancement; artificial intelligence (AI); volition; conversion

## 1. Introduction

Moral bioenhancement through AI and other technologies has the aim of improving people, making us 'better', perhaps more able to solve society's problems. This is, for example, the background to the famous Persson and Savulescu approach to moral enhancement: we should use every means we have to solve current problems, especially climate changes that threaten destruction to our planet and future generations. Their theory suggests that not enough of us have the moral capacity to react to this and other serious situations with the urgency required, and that moral bioenhancement, even involuntarily, is needed.

The hope of moral bioenhancement is that people will be able to reason better morally, not just as a good end in itself but also to spark the realization that concrete action and the will to change situations are necessary. These steps are needed for traditional moral and religious conversion, and it is proposed here that cognitive enhancement promised by the use of AI or other means could be a first step in bringing about moral enhancement. The use of AI would therefore be important in sparking or short-circuiting conversion, depending on whether or not it is able to help bring about better moral thinking as a precursor to the changed behaviour that conversion entails. Some of the challenges to moral bioenhancement as it relates to moral or religious conversion will be discussed.

## 2. AI and Cognitive Enhancement

Using AI for human enhancement has proved a great aid in restoring physical capacity. Methods used so far include deep brain stimulation (DBS), computer to brain interface, and brain implants to achieve superior learning. Several studies show some of these methods help people with reduced capacity brought about by illness or accidents, while others show the possibility of learning to operate mechanisms through neural activity, perhaps by implanted chips in the brain, thereby opening or developing neural pathways with greater capacity for cognition. Important here is the possibility of not simply being able to receive

more knowledge, but also the capacity for understanding. In other words, " . . . cognitive abilities relate to mechanisms of how we learn, remember, problem-solve and pay attention rather than with actual knowledge" (Kaimara et al. 2020).

In a recent pilot study, researchers from the National University of Singapore (NUS) showed that an artificial intelligence (AI) platform, CURATE.AI, produces training programs personalized to the individual's learning capacity to enhance training for maximum benefit. Results of the study showed the CURATE.AI platform has potential to enhance learning capacity and could lead to successful use in digital therapy, perhaps even preventing cognitive decline. Digital therapeutics using personalised applications already exist in many platforms and would be accessible to anyone with a smart phone, tablet, etc., with the potential of replacing some drug therapies and perhaps even preventing cognitive decline.

Participants' scores varied, leading to a statement by one of the authors to a daily science journal: "We need a strategy that adjusts the training—which can involve many tasks that interfere with each other—according to the participant's changing responses" (*Science Daily* 2019). It is recognized that it is difficult to standardize anything in educational theory and this remains problematic, but at the same time, personalized programs could add moral and religious content to applications (apps) tailored to the individual and helpful for cognitive and moral thinking and reasoning.

In an extensive survey on methods of cognitive enhancement conducted by the US National Institutes of Health, several different methods are discussed, noting the challenges to new methods of enhancing cognition, including the possibility of brain hacking. Dresler et al. write: "Just like the hacking culture in the realm of computer software and hardware, an increasing number of individuals experiment with strategies to creatively overcome the natural limitations of human cognitive capacity—in other words, to hack brain function" (Dresler et al. 2019). The authors note that those in the field are concerned about the usefulness of enhancement techniques when it could be employed and exploited nefariously (Dresler et al. 2019). These differing viewpoints and warnings cause hesitancy in developing open techniques using technology, while a lack of solid evidence of successful results leaves observers with questions.

The article points out that another set of disagreements arises when it is not accepted that cognitive enhancement is true enhancement, and the authors themselves demand higher standards for the use of enhancing equipment including that employing AI, etc., saying: " . . . only on the basis of a clear picture on how a particular enhancement strategy might affect specific cognitive processes in specific populations, along with side effects and costs to be expected, can an informed theoretical debate evolve and a promising empirical research designs to test the strategy can be proposed" (Dresler et al. 2019).

At the same time, the mode of action of AI in cognitive enhancement recognizes that there has been solid progress in pharmacological ways of enhancement or in behavioural intervention treatments. The NIH refers to a cluster of physical strategies for cognitive enhancement, including brain stimulation technologies. Quite apart from treatment of subjects with pathological conditions, it states, " . . . several forms of allegedly non-invasive stimulation strategies are increasingly used on healthy subjects, among them electrical stimulation methods such transcranial direct current stimulation (tDCS), transcranial alternating current stimulation (tACS), transcranial random noise stimulation (tRNS), transcranial pulsed current stimulation (tPCS), transcutaneous vagus nerve stimulation (tVNS), or median nerve stimulation (MNS)" (Dresler et al. 2019). While the authors raise doubts about the effectiveness of many of these procedures, they add a more positive note in listing 'transcranial magnetic stimulation (TMS), optical stimulation with lasers, and several forms of acoustic stimulation, such as transcranial focused ultrasound stimulation, binaural beats, or auditory stimulation of the EEG theta rhythm or sleep EEG slow oscillations' as having potential for cognitive enhancement (Dresler et al. 2019). Recently, fMRI neurofeedback is also showing potential to increase sustained attention (i.e., helpful for those with attention deficit disorders) or visuospatial memory (helpful for those with dementia) (Dresler et al. 2019).

Many of these methods function with AI assistance, and a step further in the use of AI is found in developments that the authors say 'converge minds and machines where machines are closely integrated with the person through the use of wearable electronic memory aids, AI related reality gadgets or, more permanently, bodily implants.' (Dresler et al. 2019). Neural implants that could aid memory are being tested and some are in use while brain–computer interfaces, such as those developed by Kevin Warwick, connect the central nervous system with computers through wearable or implanted electrodes to bring about enhanced cognitive function.

Indications of further use of AI in cognitive enhancement is found in commercial video games and in customized computer training programs designed to enhance specific cognitive capacities and skills. Unfortunately, recent controlled studies and meta-analyses have shed some doubt on the success of computerized brain training programs, since no single cognitive enhancer augments every cognitive function. (Dresler et al. 2019). In fact, the authors found that some cognitive training programs do enhance memory, processing speed and visuospatial skills, but work against functions and attention (Dresler et al. 2019). If an enhancement program promotes some aspects of cognition but damages others, then it will not be worth using.

For example, the studies show that electrical stimulation of posterior brain regions was found to facilitate numerical learning, whereas automaticity for the learned material was impaired. In contrast, stimulation on frontal brain regions impaired the learning process, whereas automaticity for the learned material was enhanced. Brain stimulation has thus been suggested to be a zero-sum game, with costs in some cognitive functions always being paid for gains in others (Dresler et al. 2019). This implies that enhancement may have to be tuned to the most pressing current cognitive function for the person, and certainly shows that conclusions about efficacy are rather distant, limiting not only cognitive capacity but also the capacity for moral and/or religious development through enhancement.

A major ethical and anthropological question is raised by Clowes, who notes that, "Electronic-Memory (E-Memory), powerful, portable and wearable digital gadgetry and "the cloud" of ever-present data services allow us to record, store and access an ever-expanding range of information both about and of relevance to our lives" (Clowes 2015). The cloud is the ' . . . wireless internet of data and processing services . . . which while providing local information is connected to a wireless internet that provides data-warehousing and, increasingly, processing capacities that moreover track and collect information on the minutiae of our lives' (Clowes 2015). As these technologies become more pervasive and as we grow ever more dependent on them, the author asks, " . . . but what, if anything, might be happening to our minds and sense of self as we adapt to an environment and culture increasingly populated by pervasive smart technology . . . ?" (Clowes 2015). The question is important for the possibility of cognitive and moral bioenhancement if there are negative as well as positive effects, and if, as has been shown in cognitive enhancement, there are impairment possibilities that in some ways cancel the enhanced capacities.

He is concerned about negative results on users, and questions raised by him include asking whether answers that are fed to us at the touch of a screen might dilute human capacity for thinking. We constantly use e-memory for providing information rapidly, or as an electronic diary, or, variously, as GPS/calculator/camera/video recorder of events, etc. The question is important for human capacity for learning and memory over the long haul as our dependency on machines and AI grows. On the other hand, the possibilities for cognitive enhancement, for example, in supporting the failing memory of those in the early stages of dementia, is a desirable outcome. E-memory could also have ramifications for moral bioenhancement and even for conversion, if there were good information to help people with their moral decisions. As these technologies and our habitual use of them increasingly become a part of everyday life, the tendency is for them to become invisible, fading into the background of cognition and skilled action. Clowes notes that whereas drugs that may produce cognitive enhancements or more direct brain–machine interfaces have a more public, academic and popular audience, use of the Cloud and AI is

so widespread in everyday work and tasks that we scarcely even notice our dependency (Clowes 2015).

In terms of improving cognition, he suggests e-memory provides, " . . . a scaffolding upon which we build for recall and accuracy" and this seems less threatening than the suggestion that we may be damaging human thinking, especially when he discusses how e-memory adds material we did not know before, even when we thought we 'knew' someone. E-memory adds to our store of information, and most of us are happy about that expanded knowledge and see it as positive in shaping our picture of reality, always assuming the information is accurate and verifiable, the very matters that can be problematic in this age of disinformation. Could there be cognitive diminishment in this easy access to information, even as we think our horizons are being expanded through memory aids or prompters? Will we 'learn to forget' as we become more reliant on external forces of AI for our poor memory, or will we simply use e-memory as an aid until we become familiar with the facts provided? Clowes uses the example that GPS devices guide us through areas we do not know, yet once we have navigated routes for a time, our brain takes over and we function on our own (Clowes 2015). If there is concern that our problem-solving functions and capacity for analysis could be affected, it should be remembered that e-memory is already proving valuable in helping people in cognitive decline to remember people, places and bygone times. The usual ying/yang of advantage/disadvantage applies also to technology, and time will tell if human memory will be affected by our 'not needing' to remember, e.g., telephone numbers, driving directions, historical dates, lists of capital cities, poetry or other memory lapses, now that we can even turn to portable, ever-present smart phones, tablets, wristwatches, etc., for answers.

In his article on AI as a means to moral enhancement, Klincewicz identifies a major ethical problem in noting that "There are reasons to think that leading a moral life is even more difficult today than in Aristotle's time. Many contemporary societies face rapid technological advance and moral practice is not catching up" (Klincewicz 2016). His thesis is, not unlike Persson and Savulescu's, that we are neither cognitively nor morally prepared for the advent of computers, biotechnology, and new forms of medicine. We tend to be concerned about more immediate concerns, such as family, local politics over geo-politics, and to resist action in spheres that are distant from us. Klincewicz calls this the 'Moral Lag Problem', describing all the things that cause us to be not as moral as we could or should be, and this fits with Persson and Savulescu's view that this gap threatens our planet, resulting in their urging people to take steps to remedy the problem.

He notes that Savulescu and Maslen appeal to advances in computing technology and artificial intelligence as a way of moral enhancement (Klincewicz 2016). In their view, "the moral AI would monitor physical and environmental factors that affect moral decision-making, would identify and make agents aware of their biases, and would advise agents on the right course of action, based on the agent's moral values" (Klincewicz 2016). Noting that there are concrete examples of the way in which this could be achieved, Klincewicz concludes that the approach with most promise would be to use discoveries from machine ethics along with engineering solutions featuring AI to formulate such programs to bring about moral bioenhancement (Klincewicz 2016). These ideas involve developing moral environment monitors that would prompt information about environment issues that would then 'nudge' a person towards moral conclusions to assist the person, but not attempt a take-over of the person's moral agency. Klincewicz foresees machines that would give answers to normative questions, but there could be challenges: What if I do not agree with a suggested course of action, e.g., to stop driving my car or to buy only local produce? Since machines rely on algorithms, would they not then produce a type of utilitarian ethic, for example, suggesting an answer that the greatest number of people have so far expressed? There is the possibility that the person would listen to AI suggestions over their own beliefs, since there is evidence that people can be persuaded to change their behaviour by appropriately designed technologies. Agent computer trust can be high when it comes to automation, but the problem is that humans may end up trusting an automated



system when it is not really appropriate to do so, since the machine may contain skewed information or has systemic problems of which the users are unaware. Klincewicz points to research that shows that there is reason to think that a machine that can advise and give reasons would be more successful in changing behaviour than the kind of training programs proposed by others, showing that there may be potential for creating an artificial moral advisor with AI playing a normative role (Klincewicz 2016). He notes that, "The key problem is that all of the component parts of moral AI are tied up with the agent's own moral values and those values might be based on morally compromising biases and beliefs" (Klincewicz 2016).

He suggests that a response to these challenges could lie in the authors of programs employing a morally pluralistic approach (not relativism) and points out that 'common human morality', while not always in agreement on finer points, does require some objective standards (Klincewicz 2016). I see this as an interesting referral to the possibility of some norms being seen as necessary, in contrast to today's tendency towards individual relativism or other theories that challenge the existence of any universal, objective norms. Perhaps some actions that benefit the common good or other universals, such as 'You shall not kill an innocent party', 'you shall not steal', and 'you shall not commit adultery', carry more weight than is often realized. Some also argue that any 'interference' or 'nudge' by any form of AI should allow for what Harris calls 'the freedom to fall', meaning one must decide for oneself, rightly or wrongly, and not by others' standards (Harris 2011). While against any form of compulsion in the use of AI, Klincewicz makes a good point in saying that perhaps the best points in AI's favour if used as a moral enhancer/advisor is that it " . . . invites its user to engage in rational deliberation that he or she may not have gone through otherwise" (Klincewicz 2016). After all, in any ethical theory, full information is essential for good moral decision making, and is not always easy to find.

### 3. Moral Bioenhancement

Since studies show that cognitive improvement through the use of AI is sometimes possible, the next step is to ask whether moral thinking can be enhanced by it. One view emphasizes the need for the exercise of personal moral agency by an individual, free of compulsion or manipulation. The person needs cognitive and moral capacity to sift through information and possibilities and to reflect on outcomes in order to make a freely willed moral decision. Schaefer suggests, following Jotterand, that moral neuroenhancement is impossible because, " . . . we can only become better through careful, reflective exercise of our moral agency, not through neural implementation" (Schaefer 2011). He notes a deeper problem alluded to by Jotterand: disagreement about the goal of moral enhancement threatens to make such projects untenable. I think his view is more accurately about the means used, since he asks, " . . . if part of being virtuous is to adequately process relevant factors in moral decision making, why couldn't we (at least in theory) use neural manipulation to enhance cognitive capacities and thereby make people more likely to be virtuous?" (Schaefer 2011). Jotterand, however, believes that when the word 'manipulation' is used, there is already an ethical objection: a threat to human agency and free will, an imposition of someone else's thinking on the individual concerned (Jotterand 2014).

Schaefer suggests that certain forms of cognitive manipulation would not pose the same risks to agency that, for example, emotional manipulation does. The latter could diminish agency by promoting the manipulator's values, whereas this does not necessarily happen in cognitive enhancement. He thinks the manipulator could be 'content-neutral' about values, only trying to improve the other person's ability to reason. I do not think neutrality is possible, as so many have attested to. Machine learning, in particular, has shown how algorithmic results can be skewed because of bias of various kinds, often depending on the participants featured in studies It is omnipresent and it is almost impossible for humans to be value free, a complication being that we are often unaware of our own biases. It is the hall mark of human agency that the person be free from manipulation (at least obvious manipulation!) and, therefore, free from other people's biases, in forming beliefs

and deciding on actions. Allowing for these challenges, Schaefer thinks that cognitive manipulation could make the decision-making process, including moral decision making, easier and would allow moral bioenhancement.

Jotterand acknowledges that there is strong evidence of the possibility to alter, manipulate, and regulate moral emotions using neurotechnologies or psychopharmacology. For example, increased levels of oxytocin make people more trusting and selective serotonin reuptake inhibitors (SSRIs) reduce aggression and enable cooperation (Jotterand 2014, p. 2). Similarly, the use of neurostimulation techniques seems able to produce changes in mood, affect, and moral behaviour. She accepts that these technologies can alter how people react to situations that implicate a particular moral stance. Her critique is that manipulative control of behaviour is not enough to show genuine moral enhancement, whereby the individual's moral thinking would change and develop across the spectrum. Rather, people develop morally "...through the development of a vision of the good life and an understanding of the meaning of human flourishing" (Jotterand 2011, p. 8). This is essentially Aristotelian and the accepted teaching of Thomas Aquinas, constituting my own leaning in this field but in this case does not, to me, preclude safe voluntary methods that may aid cognition and possibly moral decision making.

Regarding the question he asks about the goal of moral bioenhancement, Schaefer accepts Alasdair MacIntyre's critique that tradition-free approaches to ethics such as consequentialism and deontology, have failed to produce uncontroversial or unproblematic results, applies equally to the tradition-infused approach of virtue ethics (Schaefer 2011). If the disagreement about what it is to be good or moral remains unresolved, what is the real point of moral enhancement at all? I believe this is a valid point. Without agreement on at least some moral values and implications, the responses, even if moral bioenhancement were effective, would still leave divisions and hesitance about how to rank ethical problems in terms of priority, not to mention solutions to them. To me, this is a fundamental problem about ethics and is an ongoing dilemma in moral philosophy and theology. While we may be able to reach a degree of overlapping consensus in a few cases in the field of neuroethics (e.g., mitigation of psychopathic tendencies counts as a cognitive and moral enhancement), much of the time there may be deep moral disagreement. What might seem to be moral improvement to some could well seem moral deterioration to others and we would be divided about the content of treatments and programs. Jotterand is sceptical about consensus in these issues, writing that, "The motivation to develop biotechnologies to enhance human capacities does not occur in a vacuum, and a particular moral stance about human nature and notions of embodiment, enhancement, and morality are at play in shaping the discourse" (Jotterand 2014). Therefore, current notions of relativism, consequentialism, utilitarianism, transhumanism, libertarianism and distrust of legitimate authority and religions, and so on are at play, as well as a deepening individual relativism, where even the social notion of the common good takes second place to 'my rights' as basic justifying factors.

This disagreement is both symptomatic of and a reason for deep uncertainty about what it is to be good or moral. Various competing theories are all compelling in their own way, making adjudication of what counts as enhancement even more difficult than adjudicating the morality of actions. This makes an inclusive sort of pluralism about value attractive, but then the problem manifests itself in a different way: if, for example, we agree that being virtuous, doing the right thing and seeking the best consequences are important in being good, how do we weigh our *different* conclusions about what is right or wrong in given situations. Should majority decisions win the day, thus turning to utilitarian or pragmatic approaches? MacIntyre's critique still stands to be answered, unlikely in today's exceedingly pluralistic yet individually relativistic moral world, but answers still have to be looked for in the field of moral bioenhancement as in any other, and 'agreement to disagree' is already an answer of sorts in at least democratic societies.

Concerns about volition and privacy of thought and feelings are raised by neuroethicists such as Lavazza, who realizes the possible danger to personal freedom in applied

technology aided by AI. He notes that there are already neural prostheses " . . . depriving individuals of full control of their thoughts" (Lavazza 2018). Those who insist on the importance of the capacity for free will as a basic marker of human identity will want to safeguard those areas out of respect for human dignity and to make sure others do not acquire the right to invade private territory or to use any information obtained from patients who are treated by these means. It is easy to see how technologies could be used nefariously, but as long as most applications are used in health care treatments such as for neurodegenerative diseases, he sees no reason to think about forbidding use, since safety and cure of disease are ethical duties of the first rank.

Nonetheless, Lavazza proposes strict internal controls on what can be 'sparked' in a person's thoughts, and what can be used thereafter. He reminds us that neuroscientific techniques can be invasive, threatening a patient's cognitive freedom and privacy and therefore protective human rights have become necessary (Lavazza 2018). Access to a person's thoughts should be strictly regulated and dependent on the person's full consent to any use of material obtained. He notes this approach is necessary not only for AI devices but should become a general 'technical' operating principle to be observed by any systems connected in decoding a patient's brain activity (Lavazza 2018). I agree with this approach, and would point out another concern that there is generally a lack of enforceable regulations in many technological and health care-related fields, e.g., gene editing, because of a lack of agreement on fundamental principles. In some cases, suggested principles fail in favour of pragmatism, which could lead to severe risks to human dignity, free will and personal privacy in some instances of moral bioenhancement.

Rakic agrees with moral bioenhancement as long as it is safe and voluntary (Rakic 2017). I agree with both conditions and with his stating, like Jotterand, that, "To make it obligatory deprives us of our freedom" (Rakic 2017). He sees compulsory moral enhancement as a contradiction in terms, violating free will, and he asks "MacIntyre-type" questions such as "Whose means? Who creates the input for the 'software?' Where does thee moral authority to enhance come from? Under what terms?". He is concerned that use of any mechanism might actually reduce our will power, thus also reducing freedom of thought. He takes a hard look at the future in this perceptive statement: " . . . if such form of ultimate harm changes our species beyond recognition, compulsory moral enhancement itself obliterates humans and is, therefore, not even consonant with biological morality as an ethics of survival of the species . . . " (Rakic 2017). He finds resorting to majority decisions about these matters (which I term pragmatic rather than ethical) make matters more political than moral, because then it seems that only numbers count in ethical decision making, which for him and others is an insurmountable difficulty.

Parker Crutchfield suggests that the manipulation of a person's moral traits, i.e., the core of a person's identity, amounts to 'killing' the person (Crutchfield 2018). While the widespread use of AI or other means to perform such manipulation is still rare and mainly used as therapy, he, like Jotterand, Rakic and Lavazza, warns that we should anticipate future problems in such use and be prepared. His thesis is that change brought about by technical interventions may result in a person acting like a different person and, further, the change is not due to the person's own agency, even if voluntary. What comes to mind is Jack Nicholson's portrayal of a person changed by a lobotomy procedure (for suspicious reasons) in the movie, "One Flew over the Cuckoo's Nest", which, while fictional, resulted in a complete change in the character's identity, personality and behaviour. This is not to suggest that current brain manipulation could or would effect that level of traumatic damage, where the cure is worse than the disease, but it is a stark reminder that human manipulation could go wrong or be performed for the wrong reasons, perhaps at a cost for some.

## 4. Moral or Religious Conversion and Bioenhancement

At another level, the 'change' in moral thinking hoped for by those who advocate the use of AI in moral bioenhancement is something I shall compare with moral 'change' or



conversion, partly from a secular and partly from a Christian viewpoint. Using a popular literary example, Scrooge in Dickens' *A Christmas Carol* reveals what is meant by a spiritual or religious conversion, brought about seemingly by the 'Spirits of Christmas' in the tale. The story uses his own memories, imaginings, dreams and fear of untimely death, resulting in a late-life recognition of his own earlier woundedness and loneliness, complicated by the development of his antisocial, 'closed in' and miserly character. After Scrooge's change in heart (conversion), only Tiny Tim is allowed to invoke God expressly, but it is clear that a spiritual concept of 'love of neighbour' prevails, and Scrooge becomes a different person towards his fellow creatures. He acknowledges his earlier suffering and mistakes, he expresses repentance for those whom he had wronged, and he manifests a truly Christian spirit in making amends. These reference points are generally reckoned to be necessary for religious conversion, meaning that change in one's moral thinking leads to an actual change in behaviour. No matter what causes moral change, it is necessary for true religious conversion, and if moral bioenhancement cannot effect such change, it is more or less pointless. Crutchfield confirms this in writing, " . . . people undergo changes to their moral traits all the time, but usually these trait changes don't result in different identities because only very few traits change or because the changes occur within the person's narrative in a way that allows the narrative to continue to unify the self, preserving the person's identity through the change" (Crutchfield 2018). He is doubtful about moral bioenhancement's capacity for actual change in the person. *A Christmas Carol* is only a morality tale and may not stand up to Crutchfield's charge about real change, but it does seem to have had a great deal of influence on how people think and act and could be considered a 'universal' in capturing certain aspects of human nature, almost in the same way as a parable.

Crutchfield is concerned that if and when a change in identity occurs through bioenhancement, the person 'dies'. His concern is perhaps justified if the change is for the worse, but Saint Paul's example points to the possibility of another type of 'dying where the person is then 'reborn', and is thankful to God for that rebirth. Of course, if a person's identity is changed through external means and his or her free will is taken over by human design with the intention that he or she 'die' through bringing about radical change in personality, as happened to Jack Nicholson's character in the movie, few people would find that ethical.

Yet the possibility that the change might be positive should also be recognized. Although Saul was clearly not bioenhanced technologically, the biblical story tells us that his 'sight' was affected for some time before being restored when he underwent a drastic change in identity in becoming Paul, the follower of Christ. His conversion seems to have come from a more internal mechanism of insight and openness to grace: being 'knocked off his horse' is variously interpreted as a Scriptural way of saying that a great insight dawned on him, and he acted accordingly. He 'died' but was reborn. It can be hard for those telling their conversion stories to explain their subjective moments of insight and 'dawning' realizations, many giving witness to dramatic stories and others experiencing conversion, such as Elijah, 'in the gentle breeze'. The saying, "The bigger they come, the harder they fall", may have had some significance in the account of Saul's conversion, given his forceful and zealous nature. The main point is that many people point to the reality of conversion, and if cognitive and moral bioenhancement can set people on that path, safely and voluntarily, such enhancement could also serve a religious purpose.

## 5. Challenges to Moral or Religious Bioenhancement

Many philosophers, ethicists and scientists, however, say that evidence for effective cognitive and, in turn, moral enhancement by any means is not yet strong enough. I agree with those who say that the cognitive capacity for change in moral thinking needs to be high—especially for 'new' questions. Thinking through values and moral stances can be a difficult and ongoing task even for those in the field, often taking considerable time to arrive at conclusions or workable solutions. At the same time, another challenge exists in that new, ethical questions will always run ahead of us as technology develops at great speed and we will always be running to catch up, often *post factum*. I believe that partly

explains Persson and Savulescu's frustration, as well as Klincewicz' 'Moral Lag': if society cannot think fast and well enough to grasp the impact of a given moral dilemma, then why not enhance society to do so? Even were that to be a reality, we would still have the challenge of 'running to catch up', since it is impossible to anticipate the many questions science, medicine or technology throw our way.

Another challenge, already mentioned, is that ethicists disagree about many matters, not only on account of religion, but through disagreement about facts, sources of facts, values and norms. Assuming that is the case, we will never be sure what morally bioenhanced people will value after treatment. Short of piping the 'manipulator's' values and information into them, people are still going to think for themselves. Unless somehow 'enslaved' through a sci-fi brain–computer interface or chip (to date more sci-fi than actual), the voluntary and free will aspects of morality will be maintained. Interestingly, these are perhaps the aspects of the moral life about which most people agree. Prospects of having moral information 'piped' into a person raise the usual questions: whose information and whose morality? A lack of agreement on universal norms may still render the process of moral bioenhancement problematic at least from the standpoint of those with a specific moral agenda.

Another factor to be taken into account is that conversion is an ongoing process in spiritual life, involving cognitive, moral and religious change. It is difficult to see that compliance with, for example, Christian principles could be a *direct* result of moral bioenhancement through deep brain stimulation, e-memory or other uses of AI when human circumstances are so variable. The biblical parable of 'the sower' makes sense here: as Jesus tells it, some seed fell on rocky or stony ground, but some fell on fertile soil. Even that which fell on fertile soil was sometimes choked by weeds or was eaten by birds. Some seed grew and gave a hundred fold of itself: the same seed, but different soil and circumstances. At the religious and spiritual level, there is a need to leave space for the seed to work in us as unique individuals, and we discover that spiritual matters cannot be forced or compelled. Followers of Christianity learn that Jesus simply invites us to follow him, knowing that the harvest will not be one hundred fold. Whether moral or religious conversion occurs as a result of existing teaching methods or bioenhancement, I would venture to say that even under compulsion, the results are likely to be the same.

## 6. Conclusions

Still, cognitive enhancement is already a reality, and it looks as though it will be further developed and be more effective, at least for individuals with cognitive impairment. More people will then have improved cognition, which may in turn improve their moral thinking. We will still face disagreements about ethical theories and still run into different views of resolving those moral questions and problems. Although somewhat pessimistic, it is difficult to see that, even were it successful, moral enhancement would be able to change enough people, soon enough, to respond to more immediate global challenges such as climate change or other societal problems.

That does not mean, however, that cognitive enhancements or moral bioenhancements that are voluntary and safe are useless. If they lead to better moral thinking in individuals, they deserve a place. Better or clearer moral thinking could lead to moral and possibly religious conversion, where a person desires to change his or her behaviour, whether towards people, in choice of career, in life decisions, and so on, taking into account the values which now resonate as primary (Cf., St Paul, Scrooge—seemingly vastly different, but actually similar in experience). Moral conversion can lead us to 'see' matters in a different light and to act differently. In Canada, for example, facts have recently been revealed about the treatment of Indigenous peoples, facts that may have been deliberately concealed and obscured, while society continued in biased behaviour against these peoples, based on misinformation. When society eventually had its blinkers removed, Canada came to fully acknowledge its wrongdoing and moved to change its behaviour, a necessary

corrective in achieving justice and in allowing the process of reconciliation and healing to begin, with the intention of working towards a more egalitarian and just society.

The same process occurs in moral and religious conversion. A change in evaluative (based on cognitive) knowledge is needed first, whether sparked by spontaneous or enhanced means, leading to changed behaviour. A change in action is then needed to right the wrong that occurred, as a matter of justice. The healing process of the harm caused (sin, in religious terms) can then begin, with the person responsible for any harm resolving never to cause such harm again. Christians in Canada experienced the same moral conversion as the rest of society regarding societal treatment of Indigenous peoples over the years, but their religious convictions should have made them realize afresh how so many had abandoned or ignored their own 'Great Commandment': to love God and love one's neighbour as oneself. This does not imply love only in the affective sense, but socially and politically in accord with the important principles of social justice and the maintenance of the common good.

When it is realized how long it has taken for this wrongdoing to have been addressed, Klincewicz' point about 'moral lag' in these matters comes home to roost. Given this lag, which to future generations will appear ethically unacceptable, there is all the more reason to look for help from any sources in trying to resolve such major issues. Although there are clearly some challenges to the effectiveness of bioenhancement, advances through AI and other technologies are growing rapidly and already show potential for moral influence. With the same caveat as before as to the need for them to be safe and voluntary, their influence could be turned to great good in fostering better moral thinking and action towards achieving higher standards of individual and societal relationships.

**Funding:** This research received no external funding.

**Conflicts of Interest:** The author declares no conflict of interest.

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
