# Peer review of "Sparking Religious Conversion through AI?"

_religions, doi:10.3390/rel13050413_

Round 1
Reviewer 1 Report
The article studies AI’s role in religious conversion (both in terms of faith and action). On this account, the topic is interesting, timely. It is not only about AI, but it connects AI with a key subject of religious studies, conversion.
suggestions….
- Introduction is fine but there is too much emphasis unto climate change. One expects more direct connection with religious conversion or moral enhancement. I would suggest re-formatting the Intro. As it is, it sounds as an informal passage with some generalistic ideas.
- 29-30: The references to Jesus are in the form of typical Christian perspective. I am not criticizing this quotation but the way how it is used requires that the article will operate through a Christian epistemic framework. That quotations could be re-formatted in an indirect way. For example, “as the Christian tradition says..” That point is also valid for the article’s “Conclusion.”
- L 34. It might be better to give examples of those “several studies.” Also a very short critique of those studies (like a brief form of literature review) is required.
- L 56. The sample was quoted from ScienceDaily. Is it possible to directly refer the work if it is published elsewhere?
- L 92: “physical strategies” written in brown.
- L 131 where the article discusses Dresler’s work. The author needs to elaborate this as Dresler’s work has the potential to challenge use of AI for it presents “Brain stimulation has thus been suggested to be a zero-sum game, with costs in some cognitive functions always being paid for gains in others.” (Lines 134-135). Does the author share this argument?
- L 197: the debate on “some risks to agency” is highly useful in this regard. Is there a work that takes this problem by considering both religious/moral and medical perspectives? The crux of the issue is that “good” and “bad” in this context is both a matter of religiosity and medicine, i.e., psychology. In fact, in L 219, the author links successfully to this topic to “neuro-ethics.” So, later, the author (L 221) writes “What might seem to be moral improvement to some could well seem moral deterioration to others.” I believe that this part is crucial. Some early references (I mean in the beginning of the article) to this debate might be methodologically useful.
- L 287: The author quotes Klincewicz’s ‘machine ethics,’ and criticizes this perspective. It might be useful here to include some other scholars’ works as ‘machine ethics’ stands as a crucial element in all AI relevant procedures. What is meant by ‘machine ethics’? Can we imagine an autonomous ‘machine ethics’? This debate is also linked to what the author discusses in Line 344: If the process should be voluntary. The author agrees with Jotterand and writes that making it obligatory will deprive people of their freedom.
Author Response
Thank you for a fine review, which has proved very helpful.
1.I made a direct reference to moral enhancement /conversion in the introduction as you suggested.
2. I changed the wording of 'Jesus' comments
3. Dresler's study for the NIH was extensive and forms the better part of the background questions for cognitive enhancement as a prelude to BME. I treated their survey as a literature search for that point.
4. It was a quote by one of the authors to 'Science Daily and not part of their article
5. Gone
6. Yes, I share the concern noted by Dressler re 'zero-sum', and also state that concern as an ongoing challenge to developments in the cognitive enhancement field.
7. I think this is a crucial point for ethics in general. I inserted a point on it at the beginning, but develop it in the body of the article.
8. I agree with Klincewicz about using machine engineering in tandem with some form of 'normative 'ethics for developing AI training programs, otherwise programs would probably be devised in light of the ethical theory espoused by the programmer. If BME is the goal, use of one theory is inadequate since people are supposed to be trained to think MORE about their ethical decisions and need good and broad information. K's suggestion of some fundamental points being more 'universal' appeals to my ethical stance which leans towards the possibility of some objective norms and away from individual relativism. This part of the paper can only look briefly at this point, but I agree it is very important and possibly a permanent dilemma!
I agree with Jotterand about moral decisions being made freely and without coercion. Even legally speaking, coercion invalidates consent.
Thank you for your help!
Reviewer 2 Report
Dear author,
The research questions you present are undoubtedly significant and relevant to the research community. However, while I want to provide detailed feedback for your research, I do not think the paper is scholarly enough for publication. The manuscript failed to discuss the mechanism where religious conversion and AI can operate, both practically and theoretically. The manuscript seems to be two parts, AI and religious conversion, in different discussions without attempting to connect both. The manuscript is an opinion in nature rather than a scholarly discussion with a lack of references to both current and standing literature on bio-enhancement.
Without a clear contribution to the theory or practice, the authors cannot motivate the readers or add value to the literature from this paper. Second, I would be reluctant to use a colloquial and personal conversation style in the abstract section. Please consider deleting "I" "i.e.." Please also consider deleting "I will look at these steps in turn, and I own that my expertise in the scientific aspect of AI is limited, therefore am relying on the factual accuracy of the sources used." Authors, you are a scholar in the field. Your paper needs to have credibility for the reader to rely upon. In general, if you believe that your knowledge is limited, you must improve your knowledge until you are certain with your arguments and expertise in the field and re-write the paper. The intersecting field of AI and religion is limited, and our theoretical and training background is vastly different between the two areas, and most current papers are only explorative. However, this is not a reason for you to claim that your knowledge is limited.
Finally, your paper has high similarities to current articles, adding non-sense information about money and astronauts and failing to cite properly even in the first paragraph. Please consider reviewing information about research ethics and academic integrity.
I believe that you might be a student rather than a scholar. I like your research questions; please take my comments as constructive feedback. I hope that you can revamp the paper and I can read an excellent paper from you once a day.
Author Response
Thank you for your constructive comments!
I think there was a disconnect between AI and conversion in my paper, and I have made more links where possible. While there is material on moral bioenhancement, I found little on religious conversion. I am suggesting that moral bioenhancement is a step, but without motivation to change situations and not just experience changed moral thinking, there's a gap. I think moral conversion is sometimes a step towards experiencing religious conversion and see it's benefit both in itself and in being part of the religious process . I use several academic references to discuss moral bioenhancement, but think the next step, religious conversion tied to that, is an area for further research in this developing field and I wanted to broach that topic.
I take your points about subjective language and choice of example. Gone.
Thanks again for taking the time to respond.
Reviewer 3 Report
The article concludes with a biblical parable about the sower by which they express their reserved position on the possibility of achieving religious conversion through improvements made possible by science and technology. The value of the article is in the review of various points of view and their evaluation. In this respect, the author is clear. Their position of reservation, however, is less elaborate and remains more or less at the level of the aforementioned parable of the sower. More explicit articulation of the argument for reservation regarding the possibility of religious conversion would be welcome.
Author Response
Thank you for your helpful review!
Your point that my reservation about moral and religious conversion through AI was not explicit enough made me re-position that paragraph and, more importantly, made me think more about the many challenges to the success of moral bioenhancement in general. So yes, while expressing those reservations, I see that there are possibilities for AI programming that could 'spark' moral, and, in turn, religious conversion. That should be an interesting research topic!
Round 2
Reviewer 1 Report
I have carefully read the revised article. The author has updated the paper incorporating all of my comments. The author also provided information/responses regarding my comments. I have no other objection or suggestion.
Author Response
Thank you very much for your final review and for your advice.
I believe I have made the punctuation changes required.
Reviewer 2 Report
Dear author,
I am still quite reserved about the paper because the presentation still lacks coherence, and the contribution to the literature is not explicit. I hope that you improve your language & grammar. I can see value in your analysis, but they are mainly commentary. Please add the section on practical & theoretical contributions & suggestions for future study if applicable. I think you may want to change the title into moral enhancement with AI rather than religious conversion.
The current Turnitin result still has very high similarity with other papers. You might want to paragraph & cite more appropriately.

Author Response
No reply